# Federated Class-Heterogeneous Radiology Report Labeling with Surgical Aggregation

**Nikhil Shah**[1,2,3]                                   NIKHIL.SHAH@SOM.UMARYLAND.EDU

**Pranav Kulkarni**[1,3]  iD                              PKULKARNI@SOM.UMARYLAND.EDU

**Florence Doo**[1,2,3]  iD                               FDOO@SOM.UMARYLAND.EDU

**Ang Li**[4]                                             ANGLIECE@UMD.EDU

**Michael A. Jacobs**[5,6,7,8]  iD                        MICHAEL.A.JACOBS@UTH.TMC.EDU

**Vishwa S. Parekh**[5,9]  iD                             VISHWA.S.PAREKH@UTH.TMC.EDU

[1] *Department of Diagnostic Radiology and Nuclear Medicine, University of Maryland School of Medicine, Baltimore, MD*

[2] *University of Maryland Medical Intelligent Imaging (UM2ii) Center, University of Maryland School of Medicine, Baltimore, MD*

[3] *University of Maryland Institute for Health Computing (UM-IHC), North Bethesda, MD*

[4] *Department of Electrical and Computer Engineering, University of Maryland, College Park, MD*

[5] *Department Of Diagnostic And Interventional Imaging, McGovern Medical School, UTHealth Houston, Houston, TX*

[6] *The Russell H. Morgan Department of Radiology and Radiological Science, The Johns Hopkins University School of Medicine, Baltimore, MD*

[7] *Sidney Kimmel Comprehensive Cancer Center, The Johns Hopkins University School of Medicine, Baltimore, MD*

[8] *Department of Computer Science, Rice University, Houston, TX*

[9] *Department of Neurosurgery, The Johns Hopkins University School of Medicine, Baltimore, MD*

**Editors:** Accepted for publication at MIDL 2025

## Abstract

Labeling radiology reports is essential for creating medical imaging datasets and enabling AI-driven clinical decision support. While SBERT-based classifiers offer computationally efficient solutions for this task, a major challenge is the class heterogeneity across datasets, as different groups focus on extracting distinct disease labels. For instance, NIH and CheXpert CXR datasets share only 7 of their 14 and 13 labels, respectively. To address this, we propose to use Surgical Aggregation, a class-heterogeneous federated learning framework that collaboratively trains a global multi-label classifier without requiring alignment of labeling schemes across clients. Surgical Aggregation selectively merges shared class weights while appending new disease-specific nodes, thereby unifying distinct local labeling priorities, to dynamically incorporate all disease labels of interest. We evaluated Surgical Aggregation in multiple simulated settings with varying number of participating nodes as well as different degrees of overlapping labels. Our results demonstrate high performance confirming adaptability in class-heterogeneous environments, thereby offering a scalable and privacy-preserving solution for collaborative medical report labeling. Our code is available at https://github.com/BioIntelligence-Lab/Federated-MedEmbedX

**Keywords:** Report labeling, Label alignment, Federated learning, Chest x-ray

## 1. Introduction

The extraction of structured information from free-text radiology reports represents a fundamental task with several downstream applications, ranging from the development of AI-assisted clinical decision support systems to the optimization of clinical workflows (Doo et al., 2023; Reichenpfader et al., 2024; Savage et al., 2025). Automated labeling of radiology reports enables creation of large labeled datasets for training computer vision algorithms in detecting diseases and abnormalities. Furthermore, structured and labeled reports could also potentially facilitate automated processes such as the retrieval of patient-specific data, scheduling of follow-up appointments, and triaging of urgent cases, thereby improving diagnostic precision and patient outcomes. However, this often requires time-consuming domain expertise to navigate diverse radiology vocabularies to label radiology reports, thereby underscoring the necessity of developing a robust, scalable, and efficient medical report labeling framework.

Recent advancements in natural language processing (NLP) have positioned large language models (LLMs) as promising tools for medical report labeling (Reichenpfader et al., 2024). These models have demonstrated excellent performance across a range of use cases, including information extraction, summarization, and question answering (Mukherjee et al., 2023; Doo et al., 2024; Dorfner et al., 2024; Ma et al., 2024; Al Mohamad et al., 2025). However, LLMs face two significant limitations. First, they are computationally expensive to deploy, requiring high-end GPUs and substantial resources, which increases costs and carbon footprint. Second, LLMs are generative rather than discriminative, leading to potential issues such as failure to strictly follow instructions or consistently format responses. Finally, the performance of LLMs is significantly dependent on the quality of the prompt and requires domain expertise to curate good prompts. Thus, while LLMs offer strong performance, an ideal solution for medical report labeling should be lightweight, deterministic in performance, and have low computational requirements.

Sentence Transformers (S-BERT) offer an attractive alternative (Reimers, 2019). S-BERT is trained by fine-tuning BERT (Bidirectional Encoder Representations from Transformers) (Devlin, 2018) using a siamese network structure to capture semantic similarity between different sentences, making them ideal for downstream tasks like text labeling. A potential solution involves using S-BERT to generate embeddings of radiology report texts followed by training a lightweight multi-label classifier (e.g., MLP) to label these embeddings with different disease labels. However, the primary limitation of S-BERT based classification models is their focus on specific subsets of labels. Different research groups and institutions often prioritize extracting different sets of disease labels. For example, the NIH and CheXpert CXR datasets, two of the largest publicly available databases, contain 14 and 13 disease labels, respectively, with only 7 labels in common, as shown in Figure A.1. Consequently, S-BERT models trained on one dataset are limited in scope, lack interoperability, and result in downstream deep learning models that are suboptimal for clinical applications. This fragmentation raises the question: Can we create a global report labeling model where researchers worldwide, while developing their local labeling models? Can collaboratively build a more comprehensive medical report labeling framework?

To address this challenge, we implemented Surgical Aggregation, a federated learning framework that addresses class-heterogeneity by selectively aggregating model updates

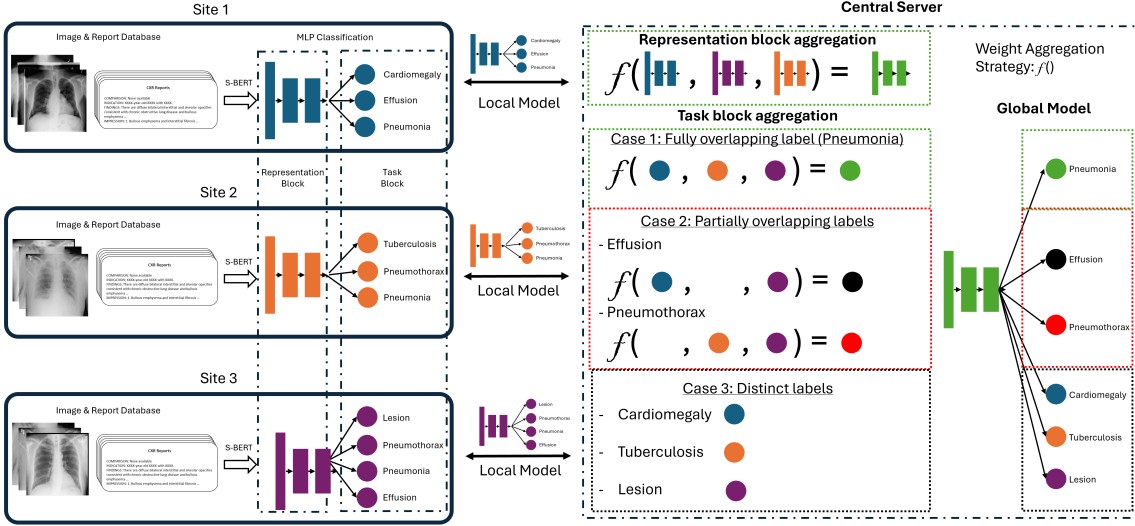

Figure 1: Surgical Aggregation based federated learning framework for class-heterogeneous multi-label chest X-ray report labeling. Local models are trained at each site using report data, with S-BERT embeddings for extracting textual features. The server aggregates representation block weights and task-specific blocks to build a global model for automatically labeling shared and distinct disease labels from distributed radiology reports.

from different clients, incorporating both shared and distinct labels without forcing label alignment (Kulkarni et al., 2025). In this work, we evaluated surgical aggregation for collaboratively labeling CXR reports across varying degrees of overlapping labels and number of participating clients.

## 2. Materials and Methods

### 2.1. Data

In this study, we used 3,665 chest radiograph reports from the Indiana University (IU) Chest X-ray Collection (Indiana Network for Patient Care), retrieved from the National Institutes of Health–National Library of Medicine's Open-i platform https://openi.nlm.nih.gov (Demner-Fushman et al., 2016). The IU dataset is a deidentified and a publicly available dataset, making it exempt from IRB review. The IU dataset reports have been annotated by board certified radiologists with thirteen predefined disease labels (CheXpert dataset (Irvin et al., 2019)). This dataset has been previously used to validate the performance of both open and closed-source LLMs for extraction of different disease labels, making it a perfect dataset for validation and comparison of the proposed technique with expert-annotated and LLM-predicted labels (Doo et al., 2024; Savage et al., 2024).

In addition, we also used a stratified subset of expert-annotated 692 radiology reports from the Medical Information Mart for Intensive Care (MIMIC) chest radiograph dataset

(Santomartino et al., 2024) MIMIC is a credentialed public access dataset consisting of radiologic images ($N = 377,110$) and reports ($N = 227,835$) collected from Beth Israel Deaconess Medical Center.

## 2.2. Radiology Report Classification Using S-BERT

### 2.2.1. S-BERT

S-BERT, or Sentence-BERT, is a variant of the BERT model fine-tuned to generate sentence embeddings that effectively preserve semantic similarity. It employs a Siamese network structure, wherein pairs of sentences are encoded using two identical BERT-based networks with shared weights. The embeddings are optimized using loss functions such as contrastive loss or triplet loss, ensuring that semantically similar sentences are mapped closer in the embedding space, while dissimilar sentences are positioned farther apart, thereby optimizing the embeddings for downstream classification and clustering tasks. For this study, we utilized the *all-MiniLM-L6-v2* S-BERT model, a lightweight variant optimized for computational efficiency.

### 2.2.2. Multi-Label Classification Using an MLP

To classify radiology reports based on S-BERT embeddings, we implemented a Multi-Layer Perceptron (MLP) as the classifier. MLPs are particularly well-suited for multi-label classification tasks, offering flexibility for modeling non-linear relationships between input features and multiple output labels. This flexibility also enables direct concantenation of new task-specific model heads - allowing seamless integration with other pre-trained models that share similar architectures, a key requirement for implementing Surgical Aggregation. The MLP model implemented in this work consisted of two fully connected hidden layers and was trained using the Adam optimizer and binary cross-entropy loss. We used the Area Under the Curve (AUC) as an evaluation metric during training.

## 2.3. Federated Learning with Surgical Aggregation

Federated learning (FL) is a decentralized paradigm that facilitates collaborative training of machine learning models while ensuring that sensitive data remains on local nodes, preserving privacy (Sandhu et al., 2023). Within this framework, we employed the surgical aggregation technique to address the challenges posed by class heterogeneity in radiology report datasets, where nodes may have distinct or partially overlapping label sets (Kulkarni et al., 2025).

In the Surgical Aggregation framework, the model architecture is divided into two components: the *representation* block and the *task* block. The representation block consists of all layers up to the final classification layer and is responsible for learning shared embeddings that capture generalizable features of the input data. The task block comprises the final classification layer, equipped with sigmoid activations, and is specialized for predicting the labels associated with each node's dataset.

The global model is constructed from scratch during aggregation, comprising a shared representation block and a task block that encompasses all tasks across participating nodes. The aggregation strategy adapts to three primary scenarios:

- **Fully Shared Labels:** When a label is present across all participating nodes, the corresponding task block weights are aggregated across nodes alongside the representation block. This ensures that knowledge pertaining to these labels is reinforced and generalized across the entire federated system.

- **Partially Shared Labels:** For labels that are shared among a subset of nodes, the task block weights corresponding to these labels are aggregated within that subset while contributing to updates in the representation block. This allows for knowledge consolidation among nodes sharing the same labels while preserving adaptability to task-specific variations.

- **Unique Labels:** When a label is exclusive to a single node, its associated task block weights are directly appended to the global model's representation block without aggregation. This approach maintains the specificity of node-exclusive tasks while ensuring seamless integration into the global framework.

In all scenarios, the weights of the representation block are aggregated across nodes to construct a robust and generalizable feature extractor. This division and aggregation strategy enable Surgical Aggregation to effectively accommodate varying levels of label overlap while maintaining scalability and interoperability across nodes. Figure 1 illustrates the complete surgical aggregation framework for collaboratively training a global model for radiology report labeling.

## 2.4. Experiments

### 2.4.1. EXPERIMENT 1: VARYING THE NUMBER OF NODES

In this experiment, we assessed how the number of nodes affects federated learning (FL) model performance. To that end, we varied the number of federated nodes from 2 to 10. In addition, to isolate the effect of node variation, the number of shared labels was fixed to 0 for all setups.

**Data Splitting:** For each nodal configuration, the dataset was divided into training and test sets with an 80-20 split. Within the training subset, the data was divided equally across all the nodes based on the number of nodes selected for each experimental configuration (ranging from 2 to 10 nodes).

**Ensuring Zero Overlap in Labels:** This dataset is multi-label, meaning a single report can contain multiple disease labels. However, for this experiment, the assigned disease labels for each node were restricted to ensure zero overlap between nodes. Once the data was split across nodes, any disease labels that were not assigned to the specific node were dropped. This represents a realistic scenario where certain diseases present in a chest X-ray report might not be reported due to the disease-specific focus of the research group.

**FL Implementation:** Federated learning was implemented by aggregating the model updates at the global server using surgical aggregation with FedAvg. The aggregation occurred after every local epoch, and this process was repeated for a total of 20 iterations. Finally, each experimental configuration was repeated 20 times with different data partitions to mitigate potential partition bias.

### 2.4.2. Experiment 2: Varying the number of shared labels in IID setting

This experiment explored the impact of shared labels by varying their number from 0 (no shared labels) to 13 (all labels shared). The number of nodes was fixed to 2 for this setup. Similar to Experiment 1, the dataset was divided into training and test sets using an 80-20 split, and the experiments were repeated 20 times to account for partition bias.

**Varying Overlap in Labels:** The number of overlapping labels between the two nodes was varied from 0 (no shared labels) to 13 (all labels shared). For each configuration, a subset of labels to be shared was randomly selected. The remaining labels were evenly distributed across the two nodes, and any non-assigned labels (apart from the shared labels) were dropped. This ensured that each node retained its distinct set of non-overlapping labels.

### 2.4.3. Experiment 3: Varying the number of shared labels in non-IID setting

This experiment focused on evaluating surgical aggregation in a realistic non-IID federated setup by varying their number from 0 (no shared labels) to 13 (all labels shared). Similar to Experiment 2, we used two nodes, but with distinct datasets—one using IU dataset reports and the other using the MIMIC reports. Unlike the IID setting, the report language as well as the distribution of non-overlapping labels differed significantly between nodes, introducing greater heterogeneity in the local training distributions. The dataset was split into training and test sets using an 80-20 ratio, and the experiment was repeated 20 times to mitigate partition bias.

### 2.4.4. Experiment 4: Comparative analysis of different embedding models

This experiment evaluates the performance of different embedding models in the federated learning framework. We assess how the choice of embeddings affects model performance by testing multiple sentence transformers and domain-specific models. The models evaluated include the S-BERT model, all-MiniLM-L6-v2, a closed-source embedding model from OpenAI (text-embedding-3-large) and a domain-specific embedding model (BioClinical BERT). The experimental setup consisted of two federated learning nodes, each with a distinct subset of data, while maintaining zero shared labels across both nodes.

### 2.4.5. Comparative models

We compared the performance of federated global models across both experiments with three baseline models.

- S-BERT with MLP classifier on the complete dataset: We trained a baseline MLP classifier for 20 epochs using S-BERT embeddings on the entire IU dataset, employing an 80-20 train-test split. To mitigate partition bias, the experiment was repeated 20 times. This model served as the upper baseline for assessing the performance of federated global models.

- LLMs: For comparison, we included the reported performance of both open- and closed-source LLMs from the literature on the same IU dataset.

2.4.6. Evaluation

We evaluated all models using accuracy for predicting each individual disease label as well as the overall accuracy. Pairwise t-tests were conducted to compare the performance of the baseline models with each federated global model, as well as to evaluate the differences between federated global models across various experimental settings. Statistical significance was defined as $p < 0.05$

## 3. Results

### 3.1. Baseline model performance

The baseline MLP classifier trained using S-BERT embeddings from the IU dataset achieved an average accuracy of $91.67\pm1.7\%$ across all disease labels on the held-out test sets across 20 iterations. In comparison, the best-performing open-source model reported in the literature is Vicuna-1.5 7B with an average zero-shot accuracy of $93.80\%$ across the complete dataset. The best-performing closed-source model in the literature, GPT-4, had an accuracy of $94.50\%$ across the entire dataset.

### 3.2. Federated learning with Surgical Aggregation

The Surgical Aggregation framework demonstrated excellent performance in tackling class heterogeneity across varying experimental configurations with different number of nodes and overlapping labels.

3.2.1. Experiment 1: Varying the Number of Nodes

As shown in Figure 2(a), the average accuracy across all federated learning configurations was significantly lower ($p < 0.05$) than the baseline performance accuracy, with the highest performance for the 2-node setup (accuracy $= 89.98 \pm 1.75\%$). Furthermore, as the number of nodes increased, performance significantly degraded ($p < 0.05$), with the lowest accuracy observed at 10 nodes ($74.73 \pm 2.29\%$). The variation in average accuracy of the surgical aggregation framework across different disease classes with changing number of nodes is illustrated in Figure 2(a). One of the potential reasons for this drop in performance could be attributed to the heterogeneity in label distribution where certain labels such Pneumothorax only have 25 positive cases (Figure 2(d)).

3.2.2. Experiment 2 and 3: Varying the Number of Shared Labels

In IID setting, there was minimal difference in the Surgical Aggregation model performance when varying the number of shared labels from 0 (no overlap) to 13 (full overlap). The highest accuracy observed was $90.89 \pm 1.4\%$, and the lowest was $90.19 \pm 1.52\%$. Figure 2(b) illustrates the performance of the surgical aggregation with varying degrees of overlapping labels demonstrating the excellent ability of surgical aggregation to deal with class heterogeneity.

We observed similar trends in the non-IID setting, where the baseline accuracy across the combined IU and MIMIC test sets was $91.15\pm0.71\%$. As in the IID setting, the performance of surgical aggregation remained relatively stable despite variations in the number of shared

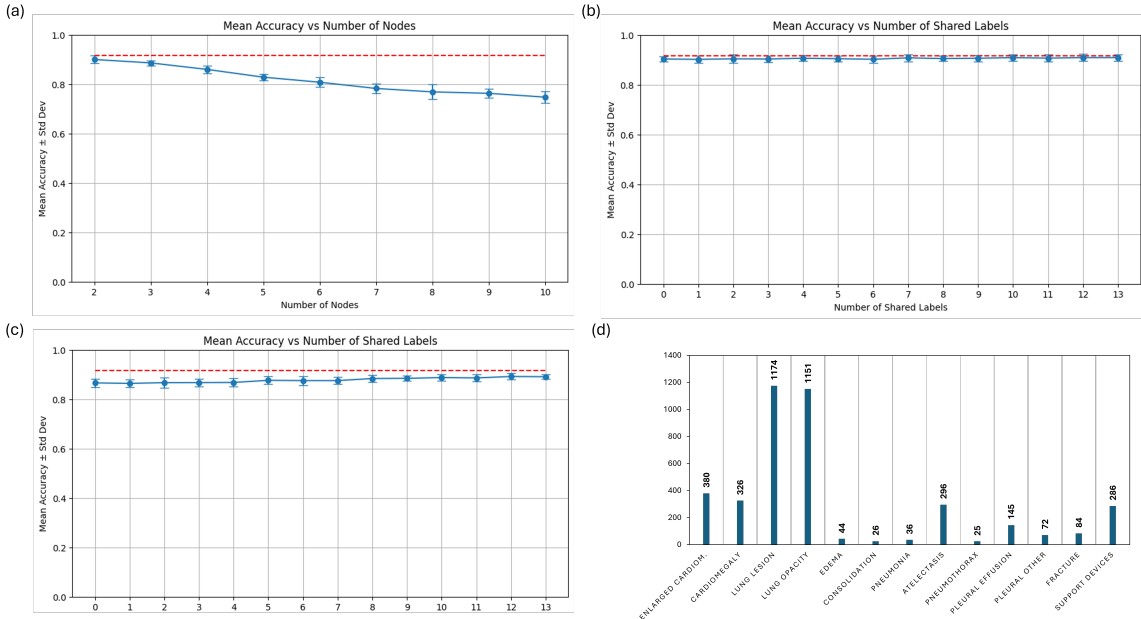

Figure 2: Comparison of average accuracy between baseline model performance (dashed red line) and surgical aggregation experiments (blue line). (a) Mean accuracy vs. Number of Nodes (b) Mean Accuracy vs. Number of Shared Labels.

labels, as shown in Figure 2(c). This demonstrates the ability of surgical aggregation to generalize effectively across class- and data-heterogeneous datasets for learning a global disease labeling model. The highest accuracy observed was $89.19 \pm 1.26\%$, and the lowest was $86.41 \pm 1.68\%$.

### 3.2.3. Experiment 4: Comparison between different embedding models

Our results indicated that the OpenAI's text-embedding-3-large model was the top performing model with an average accuracy of $93.48 \pm 1.16\%$. In comparison, all-MiniLM-v2 had an accuracy of $90.62 \pm 1.36\%$ and the clinical biobert had an accuracy of $83.82 \pm 1.90\%$. While API-based models achieve the highest accuracy, they require external data transmission, raising privacy concerns in federated learning. In contrast, domain-specific models like BioBERT can be deployed locally but exhibit lower performance, likely due to limitations in generalization. Sentence-transformers such as all-MiniLM-v2 offer a practical balance between performance, computational efficiency, and privacy, making them a viable choice for federated learning in clinical AI applications.

## 4. Discussion

The results of this study reaffirm the effectiveness of the proposed Surgical Aggregation framework in addressing the challenges of class heterogeneity in federated learning. Despite the inherent complexities of distributed learning and task variability, the framework

demonstrated its capability to aggregate local models selectively and effectively, yielding strong performance across different experimental setups. These findings underscore the potential of Surgical Aggregation as a powerful method for federated learning in multi-label classification tasks. This is especially pertinent for medical use, such as in imaging research consortia, where label overlap varies considerably across institutions.

One of the most significant advantages of Surgical Aggregation is its architectural independence (Kanhere et al., 2024; Kulkarni et al., 2025). Clients participating in the system are not required to implement any specific architecture or accommodate the tasks being undertaken by other clients. This flexibility allows for seamless integration of new nodes into the system without the need for prior knowledge of existing tasks, enabling continual system expansion. This property is especially valuable in real-world scenarios where data sources and tasks evolve over time.

The results from the overlapping labels experiment highlight the robustness of Surgical Aggregation in handling class heterogeneity. The lack of significant differences in performance across varying levels of label overlap demonstrates the framework's ability to effectively combine knowledge from heterogeneous label distributions. This result is particularly encouraging, as it suggests that Surgical Aggregation can adapt to realistic scenarios where overlapping labels are unavoidable in multi-institutional datasets.

While Surgical Aggregation performed well overall, the experiments with an increasing number of nodes revealed a decline in performance as the number of nodes increased. This trend is likely attributable to the reduction in available training samples per node as the data is distributed across more clients. Addressing this limitation may require further exploration into advanced aggregation strategies. Additionally, while Surgical Aggregation achieved strong results, the performance remains below the centralized baseline. Bridging this gap will require further refinements to the framework, validation on diverse datasets from different institutions, and application across a variety of imaging modalities and anatomical regions.

The promising initial results of this study demonstrate the potential of Surgical Aggregation as a novel federated learning framework for multi-label classification tasks. Its ability to handle class heterogeneity and scale with new nodes lays a strong foundation for future research. Expanding the framework's application to more datasets, exploring additional optimizations, and addressing the challenges identified in this study will further enhance its capabilities and establish its role as a robust solution for federated learning in diverse medical imaging scenarios.

## Acknowledgments

This research was supported in part by the UMMC/UMB Innovation Challenge Award, 2024

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

## Appendix A. Label heterogeneity across NIH and CheXpert CXR dataset

| NIH | CheXpert |
|---|---|
| Atelectasis | Atelectasis |
| Cardiomegaly | Cardiomegaly |
| Consolidation | Consolidation |
| Edema | Edema |
| Effusion | Effusion |
| Emphysema | Enlarged Cardiom. |
| Fibrosis | Fracture |
| Hernia | Lung Lesion |
| Infiltration | Lung Opacity |
| Mass | Pleural Other |
| Nodule | Pneumonia |
| Pleural Thickening | Pneuomothorax |
| Pneumonia | Support Devices |
| Pneumothorax | - |

Figure A.1: Illustration of label heterogeneity between the NIH and CheXpert datasets. Shared labels are highlighted in blue, while unique labels specific to each dataset are shown in red.

