# OpenReview forum: "Federated Class-Heterogeneous Report Labeling with Surgical Aggregation"
_MIDL.io/2025/Conference — MIDL 2025 Poster_

### Official Review · Reviewer_Fcb7 · 2025-02-17

**Confidence:** 4
**Preliminary Rating:** 4
**Final Rating:** 4

**Summary:**

This paper presents Surgical Aggregation, a novel Federated Learning (FL) framework designed to address class heterogeneity in multi-label radiology report labeling. The framework enables collaborative model training across institutions with different disease label sets without requiring label alignment. The approach aggregates shared labels while retaining distinct local labels by merging representation blocks and appending node-specific task blocks. The study uses S-BERT embeddings from the IU Chest X-ray dataset and compares Surgical Aggregation with baseline models, including S-BERT + MLP and LLMs (Vicuna-1.5, GPT-4).
Two experiments were conducted:
Experiment 1: Accuracy decreased as the number of nodes increased, with the best result for 2 nodes (89.98%) and the lowest for 10 nodes (74.73%).
Experiment 2 : Performance remained stable across different levels of label overlap, with accuracy ranging from 90.19% to 90.89%, demonstrating the framework’s robustness in handling class heterogeneity.
The results show that Surgical Aggregation effectively aggregates models in class-heterogeneous FL settings, achieving strong performance close to centralized baselines while preserving privacy.

**Strengths:**

1. The Surgical Aggregation framework effectively addresses non-overlapping label sets in FL, a key challenge in multi-institutional collaborations.
2. The method maintains high accuracy even with zero shared labels, demonstrating its flexibility for heterogeneous datasets.
3. The use of S-BERT embeddings with a compact MLP classifier offers an efficient and privacy-preserving solution for large-scale FL deployment.

**Weaknesses:**

1. Accuracy significantly drops as node count increases, highlighting the framework's scalability limitations with many participating institutions.
2. The evaluation uses only the IU Chest X-ray dataset, which limits the framework's generalizability to other modalities or multi-center datasets.

**Detailed Comments:**

1. Unrepresentative Number of Nodes: The experiment uses only 2 to 10 nodes, which is unrealistic for federated learning scenarios, where typically 100 or more clients are involved. Please justify this choice or provide experiments with a larger number of nodes to enhance practical relevance.
2. MLP Convergence Evidence: For the MLP classifier training, please include evidence that the model converged within 20 epochs, such as training loss curves or a validation accuracy plot, to demonstrate that the model was adequately trained.
3. Unexplained Performance Decline in Experiment 1: In Experiment 1, performance declines significantly from 89.98% (2 nodes) to 74.73% (10 nodes). However, the paper does not analyze the cause of this sharp drop. It is unclear whether this is due to: Data fragmentation, Aggregation inefficiencies, or Class distribution imbalance.
Please provide further investigation, such as per-class accuracy analysis, node contribution analysis, or visualizations of class distributions, to clarify why performance degrades so steeply.

**Justification Of The Final Rating:**

This paper introduces an innovative Surgical Aggregation framework for class-heterogeneous federated learning (FL) in multi-label radiology report labeling, addressing a key challenge in multi-institutional collaborations. The proposed method demonstrates strong performance, particularly in handling minimal label overlap, and provides an efficient alternative to large language models through the use of S-BERT embeddings and an MLP classifier. The authors have addressed concerns regarding dataset generalizability by incorporating a non-IID experiment with MIMIC and IU datasets and provided reasonable justifications for the limited number of clients, aligning with real-world medical FL constraints. While the unexplained performance decline in Experiment 1 remains a concern, the authors have started investigating potential causes, such as class distribution imbalance, and committed to further analysis.

**Justification Of The Preliminary Rating:**

This paper presents an innovative Surgical Aggregation framework for class-heterogeneous federated learning (FL) in multi-label radiology report labeling, addressing a significant challenge in multi-institutional collaborations. The proposed method demonstrates strong performance across experiments and maintains accuracy even with minimal label overlap, highlighting its potential for scalable, privacy-preserving model aggregation. The use of S-BERT embeddings and an MLP classifier offers an efficient solution compared to large language models, and the experiments are well-structured to explore the effects of node count and label overlap. But there are several limitations that affect the rating. Such as unrealistic number of clients, lack of convergence evidence, unexplained performance decline.

**Questions To Address In The Rebuttal:**

1. The experiment uses only 2 to 10 nodes, which is unrealistic for practical federated learning scenarios, where typically 100 or more clients are involved. Can the authors justify this choice or provide additional experiments with a larger number of nodes to demonstrate scalability and practical applicability?
2. The authors state that the MLP classifier was trained for 20 epochs, but they do not show if the model converged within this time frame. Can the authors provide training loss curves or validation accuracy plots to demonstrate that 20 epochs was sufficient for model convergence?
3. In Experiment 1, performance drops from 89.98% (2 nodes) to 74.73% (10 nodes), but the paper does not explain the cause. Can the authors investigate and clarify the reason.

---

> ### Author Response · Authors · 2025-03-08
> **Response**
>
> **1.	The evaluation uses only the IU Chest X-ray dataset, which limits the framework's generalizability to other modalities or multi-center datasets.**
>
> The reviewer raises an important point. We have included a more realistic, non-iid experiment where we conducted experiments with two non-IID nodes, one with IU dataset reports and the other with a subset of 692 MIMIC dataset reports with radiologist labels. We evaluated this across different levels of label sharing and have reported the results.
>
> **2.	Unrepresentative Number of Nodes: The experiment uses only 2 to 10 nodes, which is unrealistic for federated learning scenarios, where typically 100 or more clients are involved. Please justify this choice or provide experiments with a larger number of nodes to enhance practical relevance.**
>
> We appreciate the reviewer’s concern regarding the number of nodes in our experiments and acknowledge that large-scale federated learning (FL) deployments often involve 100 or more clients. However, in medical AI collaborations, particularly in class-heterogeneous FL, each node typically represents a hospital or institution rather than an individual device, making 2 to 10 nodes a realistic setting. Unlike standard FL scenarios with hundreds of edge devices, real-world federated collaborations in radiology often involve only a handful of participating institutions due to regulatory constraints, data-sharing agreements, and variations in labeling practices. Moreover, our non-IID experimental setup, where clients contribute different subsets of labels, requires carefully curated datasets, making it impractical to scale arbitrarily while maintaining clinical relevance. Additionally, dataset limitations and label sparsity further impact scalability; for instance, the consolidation label appears in only 26 cases, making larger-scale FL experiments difficult without introducing extreme class imbalances. While we recognize the importance of scalability, our experimental design prioritizes clinical realism and meaningful evaluation over artificially increasing the number of clients.
>
> **3.	Unexplained Performance Decline in Experiment 1: In Experiment 1, performance declines significantly from 89.98% (2 nodes) to 74.73% (10 nodes). However, the paper does not analyze the cause of this sharp drop. It is unclear whether this is due to: Data fragmentation, Aggregation inefficiencies, or Class distribution imbalance. Please provide further investigation, such as per-class accuracy analysis, node contribution analysis, or visualizations of class distributions, to clarify why performance degrades so steeply.**
>
> The reviewer raises an important question. One of the primary reasons for a sharp drop in performance could potentially be attributed to heterogeneity in the class distribution where certain labels such as pneumothorax only have 25 positive cases. This could potentially lead to scenarios where individual nodes may not have sufficient training data to train high performing models. We have added a figure illustrating the distribution of different disease labels in our dataset. However, we are evaluating other factors that the reviewer suggested and will update our response with additional metrics as we finish our analysis.

---

> > ### Comment · Reviewer_Fcb7 · 2025-03-09
> > **Response to Additional Evidence and Rating Decision**
> >
> > Thank you for providing additional evidence to address my concerns. However, I will be maintaining my current rating at this time.

---

### Official Review · Reviewer_ykDp · 2025-02-21

**Confidence:** 4
**Preliminary Rating:** 3

**Summary:**

The paper addresses the challenge of labeling radiology reports in a federated learning setting where different institutions use distinct disease labels. Traditional SBERT-based classifiers provide computationally efficient solutions but struggle with class heterogeneity, as different datasets focus on extracting different disease labels. The proposed Surgical Aggregation framework enables collaborative model training without requiring strict label alignment. This approach selectively merges shared class weights while dynamically incorporating unique disease-specific labels. Experiments on a simulated environment using the Indiana University Chest X-ray dataset demonstrate the framework’s adaptability, maintaining high classification performance across various numbers of participating nodes and levels of label overlap.

**Strengths:**

1. Clear Problem Definition and Motivation. The paper does a great job explaining why class heterogeneity in radiology report labeling is a problem, using real-world examples like the NIH and CheXpert datasets.
2. Novel Surgical Aggregation Approach. The proposed Surgical Aggregation method is a well-designed adaptation of federated learning, allowing models to collaborate without forcing label alignment. Instead of standard aggregation, it selectively merges shared class weights while keeping unique disease labels separate.

**Weaknesses:**

1. Limited Baseline Comparisons. The paper only presents SBERT-MLP with a basic FL algorithm, but it lacks benchmarks against other federated learning methods like FedProx or FedOpt. Without these, it’s unclear if the method is actually better or just different.
2. Outdated Model Choice Without Justification. The paper relies on S-BERT, which is outdated compared to LLMs (GPT, LLaMA) and CLIP. There’s no clear reason why these more advanced models weren’t tested, making the choice seem arbitrary.
3. Figure 2 Needs Better Presentation.

**Detailed Comments:**

N/A

**Justification Of The Preliminary Rating:**

The paper presents a novel approach to federated learning for radiology report labeling, but its evaluation lacks depth. The absence of comparisons with other federated learning methods (e.g., FedProx, FedOpt) makes it unclear whether the proposed Surgical Aggregation framework offers genuine advantages or is just an alternative. Additionally, the reliance on S-BERT, an outdated model, without justification, weakens the study, especially given the availability of LLMs (GPT, LLaMA) and multimodal models (CLIP) that may perform better. Lastly, Figure 2 is not well-structured, making it difficult to assess performance trends.

**Questions To Address In The Rebuttal:**

See weakness

---

> ### Author Response · Authors · 2025-03-08
> **Responses**
>
> **1.	Limited Baseline Comparisons. The paper only presents SBERT-MLP with a basic FL algorithm, but it lacks benchmarks against other federated learning methods like FedProx or FedOpt. Without these, it’s unclear if the method is actually better or just different.**
>
> The reviewer brings up an important point for comparisons with other federated learning aggregation strategies. We do want to clarify that the surgical aggregation is primarily designed to allow class-distributed or class-heterogeneous clients to collaboratively train a global model by contributing different tasks with zero or partial sharing. In contrast, techniques like FedProx are designed to address data heterogeneity in scenarios where clients have different data distributions but are still training on the same set of tasks and labels. FL techniques such as FedProx and FedOpt do not inherently support collaborative learning across clients with partial to no overlap in label sets, making it less suitable for class-heterogeneous settings. While we used FedAvg in our manuscript, Surgical Aggregation is compatible with other FL methods such as FedProx. For this reason, we limited our evaluation to surgical aggregation for addressing class-heterogeneity across different clients.
>
> **2.	Outdated Model Choice Without Justification. The paper relies on S-BERT, which is outdated compared to LLMs (GPT, LLaMA) and CLIP. There’s no clear reason why these more advanced models weren’t tested, making the choice seem arbitrary.**
>
> S-BERT models were chosen for their computational efficiency as some of these models like the one chosen in this work (all-MiniLM-L6-v2) can even run on CPUs. However, we do agree with the reviewer that the model choice needs to be justified and we have now added empirical experiments evaluating different embedding models, including clinical models like BioClinical BERT and the closed-source GPT Embedding model (text-embedding-3-large). Our results indiciate that while API-based models (text-embedding-3-large) achieve the highest accuracy, they require external data transmission, raising privacy concerns in federated learning. In contrast, domain-specific models like BioBERT can be deployed locally but exhibit lower performance, likely due to limitations in generalization. Sentence-transformers such as all-MiniLM-v2 offer a practical balance between performance, computational efficiency, and privacy, making them a viable choice for federated learning in clinical AI applications
>
> **3.	Figure 2 Needs Better Presentation**.
> We have addressed reviewer’s concern and updated Figure 2.

---

### Official Review · Reviewer_Y4dR · 2025-02-21

**Confidence:** 3
**Preliminary Rating:** 1
**Final Rating:** 3

**Summary:**

This paper presents a method called Surgical Aggregation to address the issue of label distribution discrepancies among clients in federated learning, thereby avoiding the need for label alignment. The method is experimentally validated on two chest x-ray datasets. And shows the higher performance in the class-heterogeneous environments.

**Strengths:**

- The manuscript is very well written and well organized.
- The plots demonstrating the method are clear and easy to understand.
- The research topic is important, especially considering that label misalignment among clients is very common in the medical field.

**Weaknesses:**

My primary concern lies in the similarity between this work and [1]. The referenced paper utilizes the same method, and the comparison with the baselines is not sufficiently comprehensive. Although this paper adopts a method that avoids label alignment, it is still necessary to compare with other baselines [2] [3].

* I apologize for the previous mistake; the dataset between the [1] and this paper differs. The author changed the image to the image report.

Here is the updated,

The purpose of implementing the report of the X-ray to train the model:
I believe the reason for AI4medical is to help with the clinical. However, if the report already exists, why do we need to train a model to classify it? The keyword filter is more straightforward.


The papers states

'''
In this work, we evaluated surgical aggregation for collaboratively labeling CXR reports across varying degrees of overlapping labels and number of participating clients.
'''
We got a same intuition for Fig.6 from [1]

[1]. Kulkarni, Pranav, et al. "Surgical Aggregation: Federated Class-Heterogeneous Learning." arXiv preprint arXiv:2301.06683 (2023).
[2]. Li, Tian, et al. "Federated optimization in heterogeneous networks." Proceedings of Machine learning and systems 2 (2020): 429-450.
[3]. Karimireddy, Sai Praneeth, et al. "Scaffold: Stochastic controlled averaging for federated learning." International conference on machine learning. PMLR, 2020.

**Detailed Comments:**

Please refer to the weaknesses section. Additionally, Figure 2 appears to be excessively large. If reorganized into a side-by-side structure, it could free up space for a table of results, which would enhance the readability of the paper or provide room to better explain the contributions.

**Justification Of The Final Rating:**

I appreciate this very important research topic. The application of the previous is interesting; the main concern is the simple experiment for this paper, but after more advanced language models joined, I would agree that the novel is good enough.

**Justification Of The Preliminary Rating:**

As previously mentioned, my primary concern is the similarity of this work to the earlier publication, which is my biggest worry. Considering the single-blind review process, I have noticed overlapping authorship.

After a clearer review, I acknowledge these two papers contain differences; however, changing the dataset from image to the same dataset images' report with the same method makes it hard to get a higher score.



[1]. Kulkarni, Pranav, et al. "Surgical Aggregation: Federated Class-Heterogeneous Learning." arXiv preprint arXiv:2301.06683 (2023).

**Questions To Address In The Rebuttal:**

It is necessary to further elaborate on the unique contributions of this work, considering that the method employed is directly derived from [1] and the same datasets are used.

Additionally, please include comparisons with more baselines, as relying on a single method makes it difficult to substantiate the efficacy of the proposed approach.

I would be willing to increase my rating if the contributions and experimental comparisons are enhanced.

---

> ### Author Response · Authors · 2025-03-08
> **Comment**
>
> Thank you for taking the time to provide valuable insights on our submission. We have addressed all concerns that were raised, and we feel confident that the manuscript has been improved in this process. Please see below the point-by-point response to the comments:
>
> **1.	The purpose of implementing the report of the X-ray to train the model: I believe the reason for AI4medical is to help with the clinical. However, if the report already exists, why do we need to train a model to classify it? The keyword filter is more straightforward.**
>
> The reviewer brings up an important point. Unfortunately, extracting disease labels from clinical reports is not as simple as filtering using keywords. There have been several papers in the recent years trying to tackle this problem through various algorithmic approaches, more recently, using LLMs [1-4].
>
> Please see the example from the IU dataset below illustrating the complexity of report labeling:
>
> *“INDICATION: XXXX-year-old male, chest pain
> FINDINGS: Stable enlargement of the cardiac silhouette, stable mediastinal and hilar contours, surgical clips and CABG markers. Stable XXXX densities in the left base compatible with scarring or chronic subsegmental atelectasis. No focal alveolar consolidation, no definite pleural effusion seen. Right hilar calcifications suggest a previous granulomatous process. No typical findings of pulmonary edema.
> IMPRESSION: No acute findings”*
>
> The following labels were marked by the experts as positive: Enlarged Cardiomediastinum, lung lesion, lung opacity, atelectasis, and support devices. However, not all of these keywords are present in the report. For example, the presence of surgical clips and CABG markers indicate the presence of support devices. Similarly, “lung lesions” or “opacities” are not explicitly mentioned as keywords in the report.
>
> References
>
> [1] Wei Y, Wang X, Ong H, Zhou Y, Flanders A, Shih G, Peng Y. Enhancing disease detection in radiology reports through fine-tuning lightweight LLM on weak labels. arXiv preprint arXiv:2409.16563. 2024 Sep 25.
>
> [2] Al Mohamad F, Donle L, Dorfner F, Romanescu L, Drechsler K, Wattjes MP, Nawabi J, Makowski MR, Häntze H, Adams L, Xu L. Open-source Large Language Models can Generate Labels from Radiology Reports for Training Convolutional Neural Networks. Academic Radiology. 2025 Jan 6.
>
> [3] Dorfner FJ, Jürgensen L, Donle L, Al Mohamad F, Bodenmann TR, Cleveland MC, Busch F, Adams LC, Sato J, Schultz T, Kim AE. Comparing Commercial and Open-Source Large Language Models for Labeling Chest Radiograph Reports. Radiology. 2024 Oct 29;313(1):e241139.
>
> [4] Le Guellec B, Lefèvre A, Geay C, Shorten L, Bruge C, Hacein-Bey L, Amouyel P, Pruvo JP, Kuchcinski G, Hamroun A. Performance of an open-source large language model in extracting information from free-text radiology reports. Radiology: Artificial Intelligence. 2024 May 8;6(4):e230364.
>
> **2.	Please refer to the weaknesses section. Additionally, Figure 2 appears to be excessively large. If reorganized into a side-by-side structure, it could free up space for a table of results, which would enhance the readability of the paper or provide room to better explain the contributions.**
>
> We thank the reviewer for pointing this out. We have now reduced the size of figure 2 and reorganized it.
>
> Responses continued in the next comment.

---

> > ### Author Response · Authors · 2025-03-08
> > **Responses (Continued)**
> >
> > **3.	It is necessary to further elaborate on the unique contributions of this work, considering that the method employed is directly derived from [1] and the same datasets are used. Additionally, please include comparisons with more baselines, as relying on a single method makes it difficult to substantiate the efficacy of the proposed approach. I would be willing to increase my rating if the contributions and experimental comparisons are enhanced. After a clearer review, I acknowledge these two papers contain differences; however, changing the dataset from image to the same dataset images' report with the same method makes it hard to get a higher score.**
> >
> > The reviewer raises important questions regarding the unique contributions of this work. We do want to emphasize that this paper has been submitted as an application paper (application of surgical aggregation to collaborative class-heterogeneous report labeling). Our initial results only looked at the evaluating surgical aggregation in IID setting where we conducted simulation experiments with varying number of nodes and shared labels. While these experiments established a good foundation for evaluating surgical aggregation, we do agree with the reviewer that these experiments were not sufficient. We have now included additional experiments to address reviewer’s concerns. More specifically, we have included the following new experiments and results:
> >
> > •	We have included a more realistic, non-iid experiment where we conducted experiments with two non-IID nodes, one with IU dataset reports and the other with a subset of 692 MIMIC dataset reports with radiologist labels. We evaluated this across different levels of label sharing and have reported the results.
> >
> > •	We have evaluated other embedding models such as the clinical biobert model, and openai’s text-embedding-3-large model
> >
> > By incorporating these new non-IID experiments and additional baselines, we believe our revised manuscript presents a more comprehensive evaluation of Surgical Aggregation in radiology report labeling. We hope these enhancements address the reviewer’s concerns and demonstrate the significance of our contributions.

---

> > > ### Comment · Reviewer_Y4dR · 2025-03-08
> > > **Thank you for your detailed reply**
> > >
> > > Thank you so much for your detailed response. The new experiment makes paper contribution more unique. And I agree this method is interesting enough for further evaluation. I would increase the score to boardline.

---

> > > > ### Comment · Reviewer_Y4dR · 2025-03-08
> > > > **Reason to increase**
> > > >
> > > > - Research topic is important, unaligned label
> > > > - Interesting method application
> > > > - The extra experiment during the rebuttal process makes this paper non-trivial.

---

### Author Rebuttal · Authors · 2025-03-08

**Rebuttal:**

We have uploaded the updated manuscript with the changes highlighted in red.

**Supporting Material:**

/attachment/a1fae883f446ddf408fd1a884672146ca238adc2.pdf

---

### Meta-Review · Area_Chair_zCrK · 2025-03-22

**Recommendation:** Accept (Poster)
**Confidence:** 4

**Metareview:**

This paper received 2 borderline and 1 weak accept. The authors have carefully responded to the reviewers' comments. I suggest that the authors incorporate all comments and responses into the final version.